# U-GAT-IT: Unsupervised Generative Attentional Networks with Adaptive Layer-Instance Normalization for Image-to-Image Translation

**Junho Kim**[1,2]*, **Minjae Kim**[2], **Hyeonwoo Kang**[2], **Kwang Hee Lee**[3]†

[1]Clova AI Research, NAVER Corp, [2]NCSOFT, [3]Boeing Korea Engineering and Technology Center

jhkim.ai@navercorp.com, {minjaekim, hwkang0131}@ncsoft.com, kwanghee.lee2@boeing.com

## ABSTRACT

We propose a novel method for unsupervised image-to-image translation, which incorporates a new attention module and a new learnable normalization function in an end-to-end manner. The attention module guides our model to focus on more important regions distinguishing between source and target domains based on the attention map obtained by the auxiliary classifier. Unlike previous attention-based method which cannot handle the geometric changes between domains, our model can translate both images requiring holistic changes and images requiring large shape changes. Moreover, our new AdaLIN (Adaptive Layer-Instance Normalization) function helps our attention-guided model to flexibly control the amount of change in shape and texture by learned parameters depending on datasets. Experimental results show the superiority of the proposed method compared to the existing state-of-the-art models with a fixed network architecture and hyper-parameters. Our code and datasets are available at https://github.com/taki0112/UGATIT or https://github.com/znxlwm/UGATIT-pytorch.

## 1 INTRODUCTION

Image-to-image translation aims to learn a function that maps images within two different domains. This topic has gained a lot of attention from researchers in the fields of machine learning and computer vision because of its wide range of applications including image inpainting (Pathak et al. (2014); Iizuka et al. (2017)), super resolution (Dong et al. (2016); Kim et al. (2016)), colorization (Zhang et al. (2016; 2017)) and style transfer (Gatys et al. (2016); Huang & Belongie (2017)). When paired samples are given, the mapping model can be trained in a supervised manner using a conditional generative model (Isola et al. (2017); Li et al. (2017a); Wang et al. (2018)) or a simple regression model (Larsson et al. (2016); Long et al. (2015); Zhang et al. (2016)). In unsupervised settings where no paired data is available, multiple works (Anoosheh et al. (2018); Choi et al. (2018); Huang et al. (2018); Kim et al. (2017); Liu et al. (2017); Royer et al. (2017); Taigman et al. (2017); Yi et al. (2017); Zhu et al. (2017)) successfully have translated images using shared latent space (Liu et al. (2017)) and cycle consistency assumptions (Kim et al. (2017); Zhu et al. (2017)). These works have been further developed to handle the multi-modality of the task (Huang et al. (2018)).

Despite these advances, previous methods show performance differences depending on the amount of change in both shape and texture between domains. For example, they are successful for the style transfer tasks mapping local texture (e.g., photo2vangogh and photo2portrait) but are typically unsuccessful for image translation tasks with larger shape change (e.g., selfie2anime and cat2dog) in wild images. Therefore, the pre-processing steps such as image cropping and alignment are often required to avoid these problems by limiting the complexity of the data distributions (Huang et al. (2018); Liu et al. (2017)). In addition, existing methods such as DRIT (Lee et al. (2018)) cannot

---

*Most work was done in NCSOFT.

†corresponding author

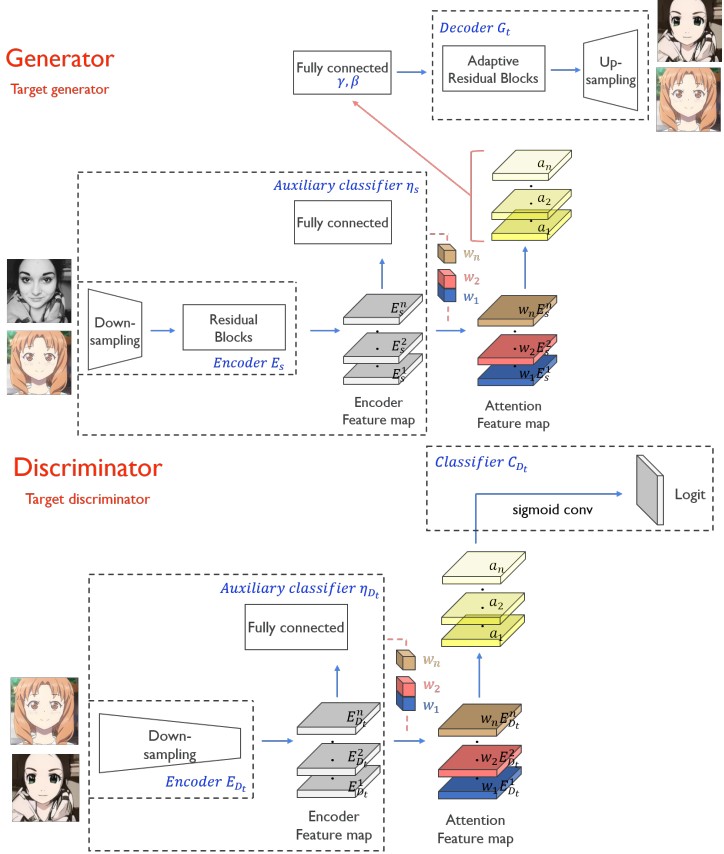

Figure 1: The model architecture of U-GAT-IT. The detailed notations are described in Section Model

acquire the desired results for both image translation preserving the shape (e.g., horse2zebra) and image translation changing the shape (e.g., cat2dog) with the fixed network architecture and hyper-parameters. The network structure or hyper-parameter setting needs to be adjusted for the specific dataset.

In this work, we propose a novel method for unsupervised image-to-image translation, which incorporates a new attention module and a new learnable normalization function in an end-to-end manner. Our model guides the translation to focus on more important regions and ignore minor regions by distinguishing between source and target domains based on the attention map obtained by the auxiliary classifier. These attention maps are embedded into the generator and discriminator to focus on semantically important areas, thus facilitating the shape transformation. While the attention map in the generator induces the focus on areas that specifically distinguish between the two domains, the attention map in the discriminator helps fine-tuning by focusing on the difference between real image and fake image in target domain. In addition to the attentional mechanism, we have found that the choice of the normalization function has a significant impact on the quality of the transformed results for various datasets with different amounts of change in shape and texture. Inspired by Batch-Instance Normalization(BIN) (Nam & Kim (2018)), we propose Adaptive Layer-Instance Normalization (AdaLIN), whose parameters are learned from datasets during training time by adaptively selecting a proper ratio between Instance normalization (IN) and Layer Normalization (LN). The AdaLIN function helps our attention-guided model to flexibly control the amount of change in shape and texture. As a result, our model, without modifying the model architecture or the hyper-parameters, can perform image translation tasks not only requiring holistic changes but also requiring large shape changes. In the experiments, we show the superiority of the proposed method compared to the existing state-of-the-art models on not only style transfer but also object transfiguration. The main contribution of the proposed work can be summarized as follows:

- We propose a novel method for unsupervised image-to-image translation with a new attention module and a new normalization function, AdaLIN.

- Our attention module helps the model to know where to transform intensively by distinguishing between source and target domains based on the attention map obtained by the auxiliary classifier.

- AdaLIN function helps our attention-guided model to flexibly control the amount of change in shape and texture without modifying the model architecture or the hyper-parameters.

## 2 UNSUPERVISED GENERATIVE ATTENTIONAL NETWORKS WITH ADAPTIVE LAYER-INSTANCE NORMALIZATION

Our goal is to train a function $G_{s \to t}$ that maps images from a source domain $X_s$ to a target domain $X_t$ using only unpaired samples drawn from each domain. Our framework consists of two generators $G_{s \to t}$ and $G_{t \to s}$ and two discriminators $D_s$ and $D_t$. We integrate the attention module into both generator and discriminator. The attention module in the discriminator guides the generator to focus on regions that are critical to generate a realistic image. The attention module in the generator gives attention to the region distinguished from the other domain. Here, we only explain $G_{s \to t}$ and $D_t$ (See Fig 1) as the vice versa should be straight-forward.

### 2.1 MODEL

#### 2.1.1 GENERATOR

Let $x \in \{X_s, X_t\}$ represent a sample from the source and the target domain. Our translation model $G_{s \to t}$ consists of an encoder $E_s$, a decoder $G_t$, and an auxiliary classifier $\eta_s$, where $\eta_s(x)$ represents the probability that $x$ comes from $X_s$. Let $E_s^k(x)$ be the $k$-th activation map of the encoder and $E_s^{k_{ij}}(x)$ be the value at $(i, j)$. Inspired by CAM (Zhou et al. (2016)), the auxiliary classifier is trained to learn the weight of the $k$-th feature map for the source domain, $w_s^k$, by using the global average pooling and global max pooling, i.e., $\eta_s(x) = \sigma(\Sigma_k w_s^k \Sigma_{ij} E_s^{k_{ij}}(x))$. By exploiting $w_s^k$, we can calculate a set of domain specific attention feature map $a_s(x) = w_s * E_s(x) = \{w_s^k * E_s^k(x) | 1 \le k \le n\}$, where $n$ is the number of encoded feature maps. Then, our translation model $G_{s \to t}$ becomes equal to $G_t(a_s(x))$. Inspired by recent works that use affine transformation parameters in normalization layers and combine normalization functions (Huang & Belongie (2017); Nam & Kim (2018)), we equip the residual blocks with AdaLIN whose parameters, $\gamma$ and $\beta$ are dynamically computed by a fully connected layer from the attention map.

$$
\begin{aligned}
AdaLIN(a, \gamma, \beta) &= \gamma \cdot (\rho \cdot \hat{a_I} + (1 - \rho) \cdot \hat{a_L}) + \beta, \\
\hat{a_I} &= \frac{a - \mu_I}{\sqrt{\sigma_I^2 + \epsilon}}, \hat{a_L} = \frac{a - \mu_L}{\sqrt{\sigma_L^2 + \epsilon}}, \\
\rho &\leftarrow clip_{[0,1]}(\rho - \tau \Delta \rho)
\end{aligned}
\tag{1}
$$

where $\mu_I$, $\mu_L$ and $\sigma_I$, $\sigma_L$ are channel-wise, layer-wise mean and standard deviation respectively, $\gamma$ and $\beta$ are parameters generated by the fully connected layer, $\tau$ is the learning rate and $\Delta \rho$ indicates the parameter update vector (e.g., the gradient) determined by the optimizer. The values of $\rho$ are constrained to the range of [0, 1] simply by imposing bounds at the parameter update step. Generator adjusts the value so that the value of $\rho$ is close to 1 in the task where the instance normalization is important and the value of $\rho$ is close to 0 in the task where the LN is important. The value of $\rho$ is initialized to 1 in the residual blocks of the decoder and 0 in the up-sampling blocks of the decoder.

An optimal method to transfer the content features onto the style features is to apply Whitening and Coloring Transform (WCT) (Li et al. (2017b)), but the computational cost is high due to the calculation of the covariance matrix and matrix inverse. Although, the AdaIN (Huang & Belongie (2017)) is much faster than the WCT, it is sub-optimal to WCT as it assumes uncorrelation between feature channels. Thus the transferred features contain slightly more patterns of the content. On the other hand, the LN (Ba et al. (2016)) does not assume uncorrelation between channels, but

sometimes it does not keep the content structure of the original domain well because it considers global statistics only for the feature maps. To overcome this, our proposed normalization technique AdaLIN combines the advantages of AdaIN and LN by selectively keeping or changing the content information, which helps to solve a wide range of image-to-image translation problems.

### 2.1.2 DISCRIMINATOR

Let $x \in \{X_t, G_{s \to t}(X_s)\}$ represent a sample from the target domain and the translated source domain. Similar to other translation models, the discriminator $D_t$ which is a multi-scale model consists of an encoder $E_{D_t}$, a classifier $C_{D_t}$, and an auxiliary classifier $\eta_{D_t}$. Unlike the other translation models, both $\eta_{D_t}(x)$ and $D_t(x)$ are trained to discriminate whether $x$ comes from $X_t$ or $G_{s \to t}(X_s)$. Given a sample $x$, $D_t(x)$ exploits the attention feature maps $a_{D_t}(x) = w_{D_t} * E_{D_t}(x)$ using $w_{D_t}$ on the encoded feature maps $E_{D_t}(x)$ that is trained by $\eta_{D_t}(x)$. Then, our discriminator $D_t(x)$ becomes equal to $C_{D_t}(a_{D_t}(x))$.

### 2.2 LOSS FUNCTION

The full objective of our model comprises four loss functions. Here, instead of using the vanilla GAN objective, we used the Least Squares GAN (Mao et al. (2017)) objective for stable training.

**Adversarial loss** An adversarial loss is employed to match the distribution of the translated images to the target image distribution:

$$L_{lsgan}^{s \to t} = (\mathbb{E}_{x \sim X_t}[(D_t(x))^2] + \mathbb{E}_{x \sim X_s}[(1 - D_t(G_{s \to t}(x)))^2]). \tag{2}$$

**Cycle loss** To alleviate the mode collapse problem, we apply a cycle consistency constraint to the generator. Given an image $x \in X_s$, after the sequential translations of $x$ from $X_s$ to $X_t$ and from $X_t$ to $X_s$, the image should be successfully translated back to the original domain:

$$L_{cycle}^{s \to t} = \mathbb{E}_{x \sim X_s}[|x - G_{t \to s}(G_{s \to t}(x)))|_1]. \tag{3}$$

**Identity loss** To ensure that the color distributions of input image and output image are similar, we apply an identity consistency constraint to the generator. Given an image $x \in X_t$, after the translation of $x$ using $G_{s \to t}$, the image should not change.

$$L_{identity}^{s \to t} = \mathbb{E}_{x \sim X_t}[|x - G_{s \to t}(x)|_1]. \tag{4}$$

**CAM loss** By exploiting the information from the auxiliary classifiers $\eta_s$ and $\eta_{D_t}$, given an image $x \in \{X_s, X_t\}$. $G_{s \to t}$ and $D_t$ get to know where they need to improve or what makes the most difference between two domains in the current state:

$$L_{cam}^{s \to t} = -(\mathbb{E}_{x \sim X_s}[log(\eta_s(x))] + \mathbb{E}_{x \sim X_t}[log(1 - \eta_s(x))]), \tag{5}$$

$$L_{cam}^{D_t} = \mathbb{E}_{x \sim X_t}[(\eta_{D_t}(x))^2] + \mathbb{E}_{x \sim X_s}[(1 - \eta_{D_t}(G_{s \to t}(x))^2]. \tag{6}$$

**Full objective** Finally, we jointly train the encoders, decoders, discriminators, and auxiliary classifiers to optimize the final objective:

$$\min_{G_{s \to t}, G_{t \to s}, \eta_s, \eta_t} \max_{D_s, D_t, \eta_{D_s}, \eta_{D_t}} \lambda_1 L_{lsgan} + \lambda_2 L_{cycle} + \lambda_3 L_{identity} + \lambda_4 L_{cam}, \tag{7}$$

where $\lambda_1 = 1, \lambda_2 = 10, \lambda_3 = 10, \lambda_4 = 1000$. Here, $L_{lsgan} = L_{lsgan}^{s \to t} + L_{lsgan}^{t \to s}$ and the other losses are defined in the similar way ($L_{cycle}$, $L_{identity}$, and $L_{cam}$)

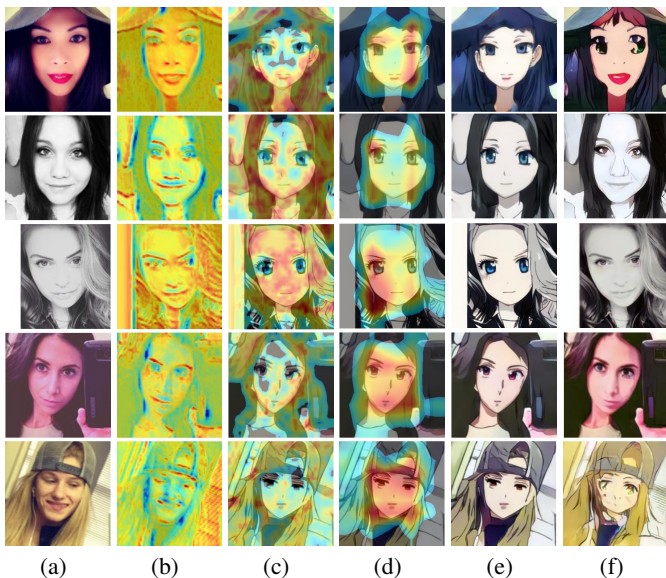

(a)       (b)       (c)       (d)       (e)       (f)

Figure 2: Visualization of the attention maps and their effects shown in the ablation experiments: (a) Source images, (b) Attention map of the generator, (c-d) Local and global attention maps of the discriminator, respectively. (e) Our results with CAM, (f) Results without CAM.

## 3 EXPERIMENTS

### 3.1 BASELINE MODEL

We have compared our method with various models including CycleGAN (Zhu et al. (2017)), UNIT (Liu et al. (2017)), MUNIT (Huang et al. (2018)), DRIT (Lee et al. (2018)), AGGAN (Mejjati et al. (2018)), and CartoonGAN (Chen et al. (2018)). All the baseline methods are implemented using the author's code.

### 3.2 DATASET

We have evaluated the performance of each method with five unpaired image datasets including four representative image translation datasets and a newly created dataset consisting of real photos and animation artworks, i.e., selfie2anime. All images are resized to 256 x 256 for training. See Appendix C for each dataset for our experiments.

### 3.3 EXPERIMENT RESULTS

We first analyze the effects of attention module and AdaLIN in the proposed model. We then compare the performance of our model against the other unsupervised image translation models listed in the previous section. To evaluate, the visual quality of translated images, we have conducted a user study. Users are asked to select the best image among the images generated from five different methods. More examples of the results comparing our model with other models are included in the supplementary materials.

#### 3.3.1 CAM ANALYSIS

First, we conduct an ablation study to confirm the benefit from the attention modules used in both generator and discriminator. As shown in Fig 2 (b), the attention feature map helps the generator to focus on the source image regions that are more discriminative from the target domain, such as eyes and mouth. Meanwhile, we can see the regions where the discriminator concentrates its attention to determine whether the target image is real or fake by visualizing local and global attention maps of

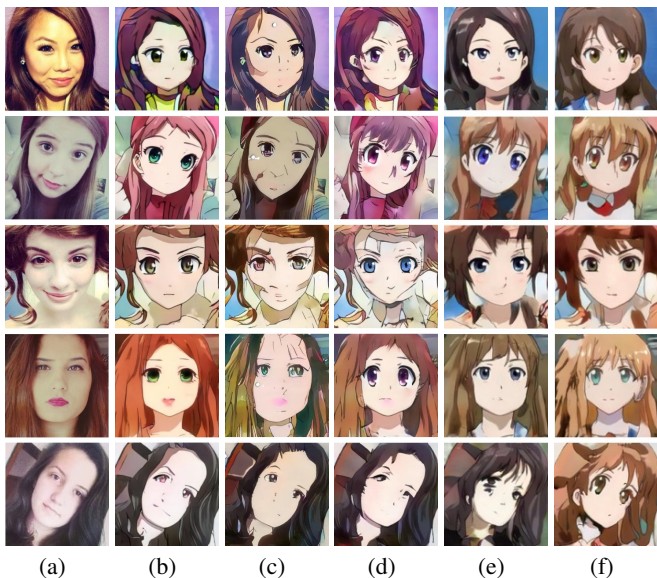

(a)         (b)         (c)         (d)         (e)         (f)

Figure 3: Comparison of the results using each normalization function: (a) Source images, (b) Our results, (c) Results only using IN in decoder with CAM, (d) Results only using LN in decoder with CAM, (e) Results only using AdaIN in decoder with CAM, (f) Results only using GN in decoder with CAM.

the discriminator as shown in Fig 2 (c) and (d), respectively. The generator can fine-tune the area where the discriminator focuses on with those attention maps. Note that we incorporate both global and local attention maps from two discriminators having different size of receptive field. Those maps can help the generator to capture the global structure (e.g., face area and near of eyes) as well as the local regions. With this information some regions are translated with more care. The results with the attention module shown in Fig 2 (e) verify the advantageous effect of exploiting attention feature map in an image translation task. On the other hand, one can see that the eyes are misaligned, or the translation is not done at all in the results without using attention module as shown in Fig 2 (f).

### 3.3.2 AdaLIN analysis

As described in Appendix B, we have applied the AdaLIN only to the decoder of the generator. The role of the residual blocks in the decoder is to embed features, and the role of the up-sampling convolution blocks in the decoder is to generate target domain images from the embedded features. If the learned value of the gate parameter $\rho$ is closer to 1, it means that the corresponding layers rely more on IN than LN. Likewise, if the learned value of $\rho$ is closer to 0, it means that the corresponding layers rely more on LN than IN. As shown in Fig 3 (c), in the case of using only IN in the decoder, the features of the source domain (e.g., earrings and shades around cheekbones) are well preserved due to channel-wise normalized feature statistics used in the residual blocks. However, the amount of translation to target domain style is somewhat insufficient since the global style cannot be captured by IN of the up-sampling convolution blocks. On the other hand, As shown in Fig 3 (d), if we use only LN in the decoder, target domain style can be transferred sufficiently by virtue of layer-wise normalized feature statistics used in the up-sampling convolution. But the features of the source domain image are less preserved by using LN in the residual blocks. This analysis of two extreme cases tells us that it is beneficial to rely more on IN than LN in the feature representation layers to preserve semantic characteristics of source domain, and the opposite is true for the up-sampling layers that actually generate images from the feature embedding. Therefore, the proposed AdaLIN which adjusts the ratio of IN and LN in the decoder according to source and target domain distributions is more preferable in unsupervised image-to-image translation tasks. Additionally, the Fig 3 (e), (f) are the results of using the AdaIN and Group Normalization (GN) (Wu & He (2018)) respectively, and our methods are showing better results compared to these.

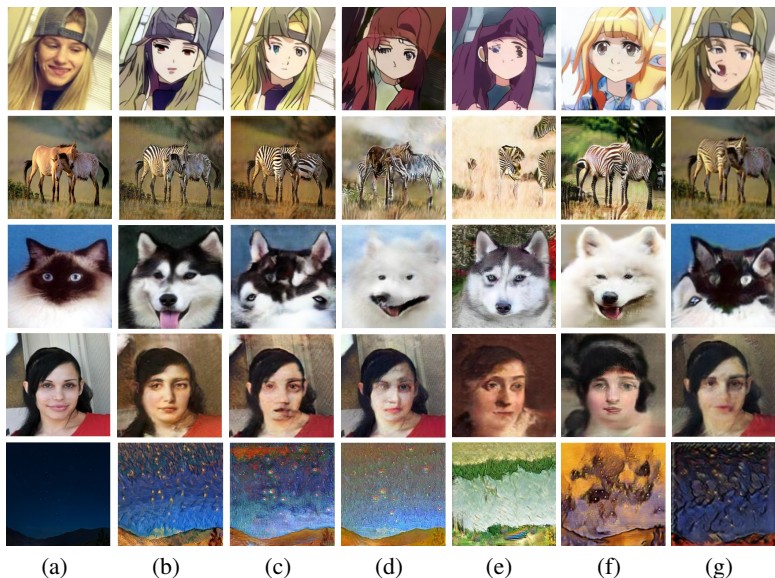

(a)    (b)    (c)    (d)    (e)    (f)    (g)

Figure 4: Visual comparisons on the five datasets. From top to bottom: selfie2anime, horse2zebra, cat2dog, photo2portrait, and photo2vangogh. (a)Source images, (b)U-GAT-IT, (c)CycleGAN, (d)UNIT, (e)MUNIT, (f)DRIT, (g)AGGAN

Table 1: Kernel Inception Distance×100±std.×100 for ablation our model. Lower is better. There are some notations; GN: Group Normalization, G_CAM: CAM of generator, D_CAM: CAM of discriminator

| Model | selfie2anime | anime2selfie |
|---|---|---|
| U-GAT-IT | $\mathbf{11.61 \pm 0.57}$ | $\mathbf{11.52 \pm 0.57}$ |
| U-GAT-IT w/ IN | $13.64 \pm 0.76$ | $13.58 \pm 0.8$ |
| U-GAT-IT w/ LN | $12.39 \pm 0.61$ | $13.17 \pm 0.8$ |
| U-GAT-IT w/ AdaIN | $12.29 \pm 0.78$ | $11.81 \pm 0.77$ |
| U-GAT-IT w/ GN | $12.76 \pm 0.64$ | $12.30 \pm 0.77$ |
| U-GAT-IT w/o CAM | $12.85 \pm 0.82$ | $14.06 \pm 0.75$ |
| U-GAT-IT w/o G_CAM | $12.33 \pm 0.68$ | $13.86 \pm 0.75$ |
| U-GAT-IT w/o D_CAM | $12.49 \pm 0.74$ | $13.33 \pm 0.89$ |

Also, as shown in Table 1, we demonstrate the performance of the attention module and AdaLIN in the selfie2anime dataset through an ablation study using Kernel Inception Distance (KID) (Bińkowski et al. (2018)). Our model achieves the lowest KID values. Even if the attention module and AdaLIN are used separately, we can see that our models perform better than the others. However, when used together, the performance is even better.

### 3.3.3 QUALITATIVE EVALUATION

For qualitative evaluation, we have also conducted a perceptual study. 135 participants are shown translated results from different methods including the proposed method with source image, and asked to select the best translated image to target domain. We inform only the name of target domain, i.e., animation, dog, and zebra to the participants. But, some example images of target domain are provided for the portrait and Van Gogh datasets as minimum information to ensure proper judgments. Table 2 shows that the proposed method achieved significantly higher score except for photo2vangogh but comparable in human perceptual study compared to other methods. In Fig 4, we present the image translation results from each method for performance comparisons. U-GAT-IT can generate undistorted image by focusing more on the distinct regions between source

Table 2: Preference score on translated images by user study.

| Model | selfie2anime | horse2zebra | cat2dog | photo2portrait | photo2vangogh |
|-------|-------------|-------------|---------|----------------|---------------|
| U-GAT-IT | **73.15** | **73.56** | **58.22** | 30.59 | **48.96** |
| CycleGAN | 20.07 | 23.07 | 6.19 | 26.59 | 27.33 |
| UNIT | 1.48 | 0.85 | 18.63 | **32.11** | 11.93 |
| MUNIT | 3.41 | 1.04 | 14.48 | 8.22 | 2.07 |
| DRIT | 1.89 | 1.48 | 2.48 | 2.48 | 9.70 |

Table 3: Kernel Inception Distance$\times 100 \pm$std.$\times 100$ for difference image translation mode. Lower is better.

| Model | selfie2anime | horse2zebra | cat2dog | photo2portrait | photo2vangogh |
|-------|-------------|-------------|---------|----------------|---------------|
| U-GAT-IT | **11.61 $\pm$ 0.57** | **7.06 $\pm$ 0.8** | **7.07 $\pm$ 0.65** | 1.79 $\pm$ 0.34 | 4.28 $\pm$ 0.33 |
| CycleGAN | 13.08 $\pm$ 0.49 | 8.05 $\pm$ 0.72 | 8.92 $\pm$ 0.69 | 1.84 $\pm$ 0.34 | 5.46 $\pm$ 0.33 |
| UNIT | 14.71 $\pm$ 0.59 | 10.44 $\pm$ 0.67 | 8.15 $\pm$ 0.48 | **1.20 $\pm$ 0.31** | **4.26 $\pm$ 0.29** |
| MUNIT | 13.85 $\pm$ 0.41 | 11.41 $\pm$ 0.83 | 10.13 $\pm$ 0.27 | 4.75 $\pm$ 0.52 | 13.08 $\pm$ 0.34 |
| DRIT | 15.08 $\pm$ 0.62 | 9.79 $\pm$ 0.62 | 10.92 $\pm$ 0.33 | 5.85 $\pm$ 0.54 | 12.65 $\pm$ 0.35 |
| AGGAN | 14.63 $\pm$ 0.55 | 7.58 $\pm$ 0.71 | 9.84 $\pm$ 0.79 | 2.33 $\pm$ 0.36 | 6.95 $\pm$ 0.33 |
| CartoonGAN | 15.85 $\pm$ 0.69 | - | - | - | - |
| **Model** | **anime2selfie** | **zebra2horse** | **dog2cat** | **portrait2photo** | **vangogh2photo** |
| U-GAT-IT | **11.52 $\pm$ 0.57** | **7.47 $\pm$ 0.71** | **8.15 $\pm$ 0.66** | 1.69 $\pm$ 0.53 | 5.61 $\pm$ 0.32 |
| CycleGAN | 11.84 $\pm$ 0.74 | 8.0 $\pm$ 0.66 | 9.94 $\pm$ 0.36 | 1.82 $\pm$ 0.36 | **4.68 $\pm$ 0.36** |
| UNIT | 26.32 $\pm$ 0.92 | 14.93 $\pm$ 0.75 | 9.81 $\pm$ 0.34 | **1.42 $\pm$ 0.24** | 9.72 $\pm$ 0.33 |
| MUNIT | 13.94 $\pm$ 0.72 | 16.47 $\pm$ 1.04 | 10.39 $\pm$ 0.25 | 3.30 $\pm$ 0.47 | 9.53 $\pm$ 0.35 |
| DRIT | 14.85 $\pm$ 0.60 | 10.98 $\pm$ 0.55 | 10.86 $\pm$ 0.24 | 4.76 $\pm$ 0.72 | 7.72 $\pm$ 0.34 |
| AGGAN | 12.72 $\pm$ 1.03 | 8.80 $\pm$ 0.66 | 9.45 $\pm$ 0.64 | 2.19 $\pm$ 0.40 | 5.85 $\pm$ 0.31 |

and target domain by exploiting the attention modules. Note that the regions around heads of two zebras or eyes of dog are distorted in the results from CycleGAN. Moreover, translated results using U-GAT-IT are visually superior to other methods while preserving semantic features of source domain. It is worth noting that the results from MUNIT and DRIT are much dissimilar to the source images since they generate images with random style codes for diversity. Furthermore, it should be emphasized that U-GAT-IT have applied with the same network architecture and hyper-parameters for all of the five different datasets, while the other algorithms are trained with preset networks or hyper-parameters. Through the results of user study, we show that the combination of our attention module and AdaLIN makes our model more flexible.

### 3.3.4 QUANTITATIVE EVALUATION

For quantitative evaluation, we use the recently proposed KID, which computes the squared Maximum Mean Discrepancy between the feature representations of real and generated images. The feature representations are extracted from the Inception network (Szegedy et al. (2016)). In contrast to the Fréchet Inception Distance (Heusel et al. (2017)), KID has an unbiased estimator, which makes it more reliable, especially when there are fewer test images than the dimensionality of the inception features. The lower KID indicates that the more shared visual similarities between real and generated images (Mejjati et al. (2018)). Therefore, if well translated, the KID will have a small value in several datasets. Table 3 shows that the proposed method achieved the lowest KID scores except for the style transfer tasks like photo2vangogh and photo2portrait. However, there is no big difference from the lowest score. Also, unlike UNIT and MUNIT, we can see that the source $\rightarrow$ target, target $\rightarrow$ source translations are both stable. U-GAT-IT shows even lower KID than the recent attention-based method, AGGAN. AGGAN yields poor performance for the transformation with shape change such as dog2cat and anime2selfie unlike the U-GAT-IT, the attention module of which focuses on distinguishing not between background and foreground but differences between

two domains. CartoonGAN, as shown in the supplementary materials, has only changed the overall color of the image to an animated style, but compared to selfie, the eye, which is the biggest characteristic of animation, has not changed at all. Therefore, CartoonGAN has the higher KID.

## 4 CONCLUSIONS

In this paper, we have proposed unsupervised image-to-image translation (U-GAT-IT), with the attention module and AdaLIN which can produce more visually pleasing results in various datasets with a fixed network architecture and hyper-parameter. Detailed analysis of various experimental results supports our assumption that attention maps obtained by an auxiliary classifier can guide generator to focus more on distinct regions between source and target domain. In addition, we have found that the Adaptive Layer-Instance Normalization (AdaLIN) is essential for translating various datasets that contains different amount of geometry and style changes. Through experiments, we have shown that the superiority of the proposed method compared to the existing state-of-the-art GAN-based models for unsupervised image-to-image translation tasks.

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

# A  RELATED WORKS

## A.1  GENERATIVE ADVERSARIAL NETWORKS

Generative Adversarial Networks (GAN)(Goodfellow et al. (2014)) have achieved impressive results on a wide variety of image generation(Arjovsky et al. (2017); Berthelot et al. (2017); Karras et al. (2018); Zhao et al. (2017)), image inpainting(Iizuka et al. (2017)), image translation(Choi et al. (2018); Huang et al. (2018); Isola et al. (2017); Liu et al. (2017); Wang et al. (2018); Zhu et al. (2017)) tasks. In training, a generator aims to generate realistic images to fool a discriminator while the discriminator tries to distinguish the generated images from real images. Various multi-stage generative models(Karras et al. (2018); Wang et al. (2018)) and better training objectives(Arjovsky et al. (2017); Berthelot et al. (2017); Mao et al. (2017); Zhao et al. (2017)) have been proposed to generate more realistic images. In this paper, our model uses GAN to learn the transformation from a source domain to a significantly different target domain, given unpaired training data.

## A.2  IMAGE-TO-IMAGE TRANSLATION

Isola et al.(Isola et al. (2017)) have proposed a conditional GAN-based unified framework for image-to-image translation. High-resolution version of the pix2pix have been proposed by Wang et al.(Wang et al. (2018)) Recently, there have been various attempts (Huang et al. (2018); Kim et al. (2017); Liu et al. (2017); Taigman et al. (2017); Zhu et al. (2017)) to learn image translation from an unpaired dataset. CycleGAN (Zhu et al. (2017)) have proposed a cyclic consistence loss for the first time to enforce one-to-one mapping. UNIT (Liu et al. (2017)) assumed a shared-latent space to tackle unsupervised image translation. However, this approach performs well only when the two domains have similar patterns. MUNIT (Huang et al. (2018)) makes it possible to extend to many-to-many mapping by decomposing the image into content code that is domain-invariant and a style code that captures domain-specific properties. MUNIT synthesizes the separated content and style to generate the final image, where the image quality is improved by using adaptive instance normalization (Huang & Belongie (2017)). With the same purpose as MUNIT, DRIT (Lee et al. (2018)) decomposes images into content and style, so that many-to-many mapping is possible. The only difference is that content space is shared between the two domains using the weight sharing and content discriminator which is auxiliary classifier. Nevertheless, the performance of these methods (Huang et al. (2018); Liu et al. (2017); Lee et al. (2018)) are limited to the dataset that contains well-aligned images between source and target domains. In addition, AGGAN (Mejjati et al. (2018)) improved the performance of image translation by using attention mechanism to distinguish between foreground and background. However, the attention module in AGGAN cannot help to transform the object's shape in the image. Although, CartoonGAN (Chen et al. (2018)) shows good performance for animation style translation, it changes only the color, tone, and thickness of line in the image. Therefore it is not suitable for the shape change in the image.

## A.3  CLASS ACTIVATION MAP

Zhou et al. (Zhou et al. (2016)) have proposed Class Activation Map (CAM) using global average pooling in a CNN. The CAM for a particular class shows the discriminative image regions by the CNN to determine that class. In this work, our model leads to intensively change discriminative image regions provided by distinguishing two domains using the CAM approach. However, not only global average pooling is used, but global max pooling is also used to make the results better.

## A.4  NORMALIZATION

Recent neural style transfer researches have shown that CNN feature statistics (e.g., Gram matrix (Gatys et al. (2016)), mean and variance (Huang & Belongie (2017)) can be used as direct descriptors for image styles. In particular, Instance Normalization (IN) has the effect of removing the style variation by directly normalizing the feature statistics of the image and is used more often than Batch Normalization (BN) or Layer Normalization (LN) in style transfer. However, when normalizing images, recent studies use Adaptive Instance Normalization (AdaIN) (Huang & Belongie (2017)), Conditional Instance Normalization (CIN) (Dumoulin et al. (2017)), and Batch-Instance Normalization (BIN) (Nam & Kim (2018)) instead of using IN alone. In our work, we propose an Adaptive

Layer-Instance Normalization (AdaLIN) function to adaptively select a proper ratio between IN and LN. Through the AdaLIN, our attention-guided model can flexibly control the amount of change in shape and texture.

# B IMPLEMENTATION DETAILS

## B.1 NETWORK ARCHITECTURE

The network architectures of U-GAT-IT are shown in Table 4, 5, and 6. The encoder of the generator is composed of two convolution layers with the stride size of two for down-sampling and four residual blocks. The decoder of the generator consists of four residual blocks and two up-sampling convolution layers with the stride size of one. Note that we use the instance normalization for the encoder and AdaLIN for the decoder, respectively. In general, LN does not perform better than batch normalization in classification problems (Wu & He (2018)). Since the auxiliary classifier is connected from the encoder in the generator, to increase the accuracy of the auxiliary classifier we use the instance normalization(batch normalization with a mini-batch size of 1) instead of the AdaLIN. Spectral normalization (Miyato et al. (2018)) is used for the discriminator. We employ two different scales of PatchGAN (Isola et al. (2017)) for the discriminator network, which classifies whether local (70 x 70) and global (286 x 286) image patches are real or fake. For the activation function, we use ReLU in the generator and leaky-ReLU with a slope of 0.2 in the discriminator.

## B.2 TRAINING

All models are trained using Adam (Kingma & Ba (2015)) with $\beta_1$=0.5 and $\beta_2$=0.999. For data augmentation, we flipped the images horizontally with a probability of 0.5, resized them to 286 x 286, and random cropped them to 256 x 256. The batch size is set to one for all experiments. We train all models with a fixed learning rate of 0.0001 until 500,000 iterations and linearly decayed up to 1,000,000 iterations. We also use a weight decay at rate of 0.0001. The weights are initialized from a zero-centered normal distribution with a standard deviation of 0.02.

# C DATASET DETAILS

**selfie2anime** The selfie dataset contains 46,836 selfie images annotated with 36 different attributes. We only use photos of females as training data and test data. The size of the training dataset is 3400, and that of the test dataset is 100, with the image size of 256 x 256. For the anime dataset, we have firstly retrieved 69,926 animation character images from Anime-Planet[1]. Among those images, 27,023 face images are extracted by using an anime-face detector[2]. After selecting only female character images and removing monochrome images manually, we have collected two datasets of female anime face images, with the sizes of 3400 and 100 for training and test data respectively, which is the same numbers as the selfie dataset. Finally, all anime face images are resized to 256 x 256 by applying a CNN-based image super-resolution algorithm[3].

**horse2zebra and photo2vangogh** These datasets are used in CycleGAN (Zhu et al. (2017)). The training dataset size of each class: 1,067 (horse), 1,334 (zebra), 6,287 (photo), and 400 (vangogh). The test datasets consist of 120 (horse), 140 (zebra), 751 (photo), and 400 (vangogh). Note that the training data and the test data of vangogh class are the same.

**cat2dog and photo2portrait** These datasets are used in DRIT (Lee et al. (2018)). The numbers of data for each class are 871 (cat), 1,364 (zebra), 6,452 (photo), and 1,811 (vangogh). We use 120 (horse), 140 (zebra), 751 (photo), and 400 (vangogh) randomly selected images as test data, respectively.

---

[1]http://www.anime-planet.com/
[2]https://github.com/nagadomi/lbpcascade_animeface
[3]https://github.com/nagadomi/waifu2x

## D ADDITIONAL EXPERIMENTAL RESULTS

In addition to the results presented in the paper, we show supplement generation results for the five datasets in Figs 5, 6, 7, 8, 9, 10, 11, and 12.

Table 4: The detail of generator architecture.

| Part | Input → Output Shape | Layer Information |
|---|---|---|
| Encoder Down-sampling | $(h, w, 3) \rightarrow (h, w, 64)$ | CONV-(N64, K7, S1, P3), IN, ReLU |
| | $(h, w, 64) \rightarrow (\frac{h}{2}, \frac{w}{2}, 128)$ | CONV-(N128, K3, S2, P1), IN, ReLU |
| | $(\frac{h}{2}, \frac{w}{2}, 128) \rightarrow (\frac{h}{4}, \frac{w}{4}, 256)$ | CONV-(N256, K3, S2, P1), IN, ReLU |
| Encoder Bottleneck | $(\frac{h}{4}, \frac{w}{4}, 256) \rightarrow (\frac{h}{4}, \frac{w}{4}, 256)$ | ResBlock-(N256, K3, S1, P1), IN, ReLU |
| | $(\frac{h}{4}, \frac{w}{4}, 256) \rightarrow (\frac{h}{4}, \frac{w}{4}, 256)$ | ResBlock-(N256, K3, S1, P1), IN, ReLU |
| | $(\frac{h}{4}, \frac{w}{4}, 256) \rightarrow (\frac{h}{4}, \frac{w}{4}, 256)$ | ResBlock-(N256, K3, S1, P1), IN, ReLU |
| | $(\frac{h}{4}, \frac{w}{4}, 256) \rightarrow (\frac{h}{4}, \frac{w}{4}, 256)$ | ResBlock-(N256, K3, S1, P1), IN, ReLU |
| CAM of Generator | $(\frac{h}{4}, \frac{w}{4}, 256) \rightarrow (\frac{h}{4}, \frac{w}{4}, 512)$ | Global Average & Max Pooling, MLP-(N1), Multiply the weights of MLP |
| | $(\frac{h}{4}, \frac{w}{4}, 512) \rightarrow (\frac{h}{4}, \frac{w}{4}, 256)$ | CONV-(N256, K1, S1), ReLU |
| $\gamma, \beta$ | $(\frac{h}{4}, \frac{w}{4}, 256) \rightarrow (1, 1, 256)$ | MLP-(N256), ReLU |
| | $(1, 1, 256) \rightarrow (1, 1, 256)$ | MLP-(N256), ReLU |
| | $(1, 1, 256) \rightarrow (1, 1, 256)$ | MLP-(N256), ReLU |
| Decoder Bottleneck | $(\frac{h}{4}, \frac{w}{4}, 256) \rightarrow (\frac{h}{4}, \frac{w}{4}, 256)$ | AdaResBlock-(N256, K3, S1, P1), AdaILN, ReLU |
| | $(\frac{h}{4}, \frac{w}{4}, 256) \rightarrow (\frac{h}{4}, \frac{w}{4}, 256)$ | AdaResBlock-(N256, K3, S1, P1), AdaILN, ReLU |
| | $(\frac{h}{4}, \frac{w}{4}, 256) \rightarrow (\frac{h}{4}, \frac{w}{4}, 256)$ | AdaResBlock-(N256, K3, S1, P1), AdaILN, ReLU |
| | $(\frac{h}{4}, \frac{w}{4}, 256) \rightarrow (\frac{h}{4}, \frac{w}{4}, 256)$ | AdaResBlock-(N256, K3, S1, P1), AdaILN, ReLU |
| Decoder Up-sampling | $(\frac{h}{4}, \frac{w}{4}, 256) \rightarrow (\frac{h}{2}, \frac{w}{2}, 128)$ | Up-CONV-(N128, K3, S1, P1), LIN, ReLU |
| | $(\frac{h}{2}, \frac{w}{2}, 128) \rightarrow (h, w, 64)$ | Up-CONV-(N64, K3, S1, P1), LIN, ReLU |
| | $(h, w, 64) \rightarrow (h, w, 3)$ | CONV-(N3, K7, S1, P3), Tanh |

Table 5: The detail of local discriminator.

| Part | Input → Output Shape | Layer Information |
|---|---|---|
| Encoder Down-sampling | $(h, w, 3) \rightarrow (\frac{h}{2}, \frac{w}{2}, 64)$ | CONV-(N64, K4, S2, P1), SN, Leaky-ReLU |
| | $(\frac{h}{2}, \frac{w}{2}, 64) \rightarrow (\frac{h}{4}, \frac{w}{4}, 128)$ | CONV-(N128, K4, S2, P1), SN, Leaky-ReLU |
| | $(\frac{h}{4}, \frac{w}{4}, 128) \rightarrow (\frac{h}{8}, \frac{w}{8}, 256)$ | CONV-(N256, K4, S2, P1), SN, Leaky-ReLU |
| | $(\frac{h}{8}, \frac{w}{8}, 256) \rightarrow (\frac{h}{8}, \frac{w}{8}, 512)$ | CONV-(N512, K4, S1, P1), SN, Leaky-ReLU |
| CAM of Discriminator | $(\frac{h}{8}, \frac{w}{8}, 512) \rightarrow (\frac{h}{8}, \frac{w}{8}, 1024)$ | Global Average & Max Pooling, MLP-(N1), Multiply the weights of MLP |
| | $(\frac{h}{8}, \frac{w}{8}, 1024) \rightarrow (\frac{h}{8}, \frac{w}{8}, 512)$ | CONV-(N512, K1, S1), Leaky-ReLU |
| Classifier | $(\frac{h}{8}, \frac{w}{8}, 512) \rightarrow (\frac{h}{8}, \frac{w}{8}, 1)$ | CONV-(N1, K4, S1, P1), SN |

Table 6: The detail of global discriminator.

| Part | Input $\to$ Output Shape | Layer Information |
|---|---|---|
| Encoder Down-sampling | $(h, w, 3) \to (\frac{h}{2}, \frac{w}{2}, 64)$ | CONV-(N64, K4, S2, P1), SN, Leaky-ReLU |
| | $(\frac{h}{2}, \frac{w}{2}, 64) \to (\frac{h}{4}, \frac{w}{4}, 128)$ | CONV-(N128, K4, S2, P1), SN, Leaky-ReLU |
| | $(\frac{h}{4}, \frac{w}{4}, 128) \to (\frac{h}{8}, \frac{w}{8}, 256)$ | CONV-(N256, K4, S2, P1), SN, Leaky-ReLU |
| | $(\frac{h}{8}, \frac{w}{8}, 256) \to (\frac{h}{16}, \frac{w}{16}, 512)$ | CONV-(N512, K4, S2, P1), SN, Leaky-ReLU |
| | $(\frac{h}{16}, \frac{w}{16}, 512) \to (\frac{h}{32}, \frac{w}{32}, 1024)$ | CONV-(N1024, K4, S2, P1), SN, Leaky-ReLU |
| | $(\frac{h}{32}, \frac{w}{32}, 1024) \to (\frac{h}{32}, \frac{w}{32}, 2048)$ | CONV-(N2048, K4, S1, P1), SN, Leaky-ReLU |
| CAM of Discriminator | $(\frac{h}{32}, \frac{w}{32}, 2048) \to (\frac{h}{32}, \frac{w}{32}, 4096)$ | Global Average & Max Pooling, MLP-(N1), Multiply the weights of MLP |
| | $(\frac{h}{32}, \frac{w}{32}, 4096) \to (\frac{h}{32}, \frac{w}{32}, 2048)$ | CONV-(N2048, K1, S1), Leaky-ReLU |
| Classifier | $(\frac{h}{32}, \frac{w}{32}, 2048) \to (\frac{h}{32}, \frac{w}{32}, 1)$ | CONV-(N1, K4, S1, P1), SN |

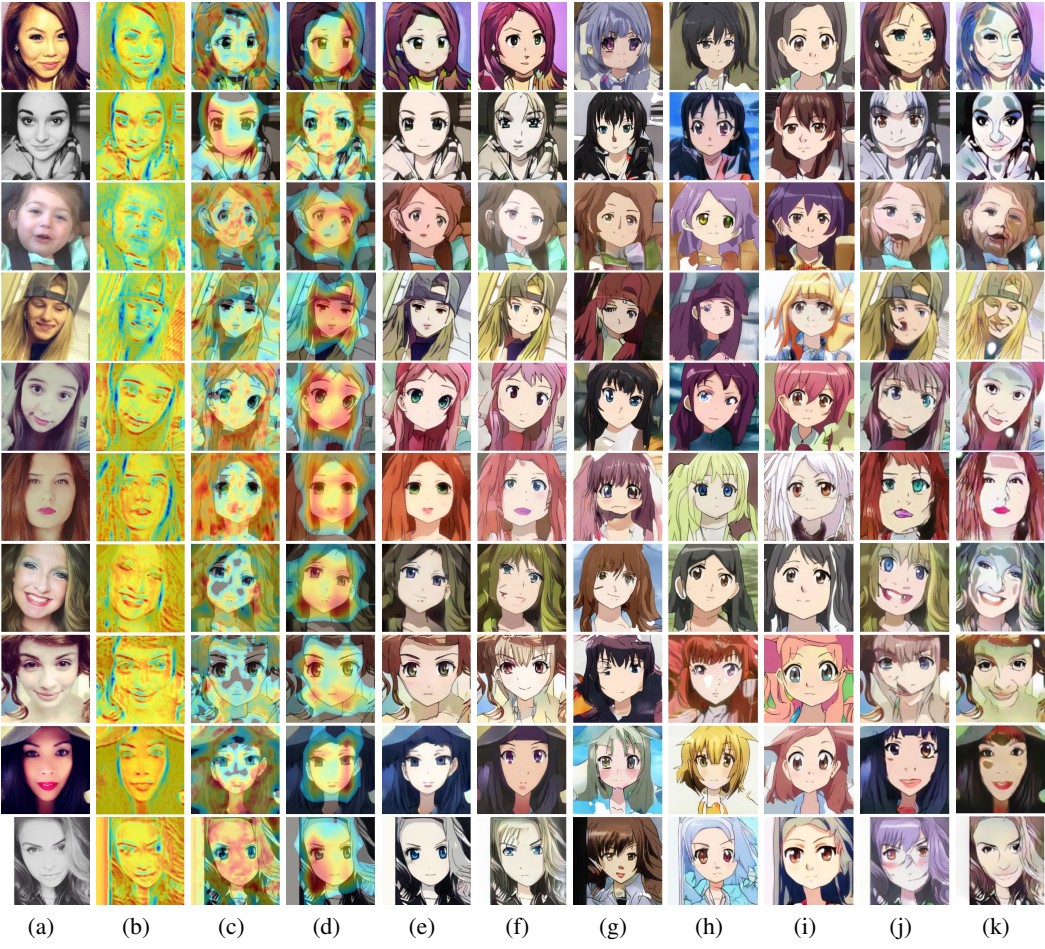

(a) (b) (c) (d) (e) (f) (g) (h) (i) (j) (k)

Figure 5: Visual comparisons of the selfie2anime with attention features maps. (a) Source images, (b) Attention map of the generator, (c-d) Local and global attention maps of the discriminators, (e) Our results, (f) CycleGAN (Zhu et al. (2017)), (g) UNIT (Liu et al. (2017)), (h) MUNIT (Huang et al. (2018)), (i) DRIT (Lee et al. (2018)), (j) AGGAN (Mejjati et al. (2018)), (k) CartoonGAN (Chen et al. (2018)).

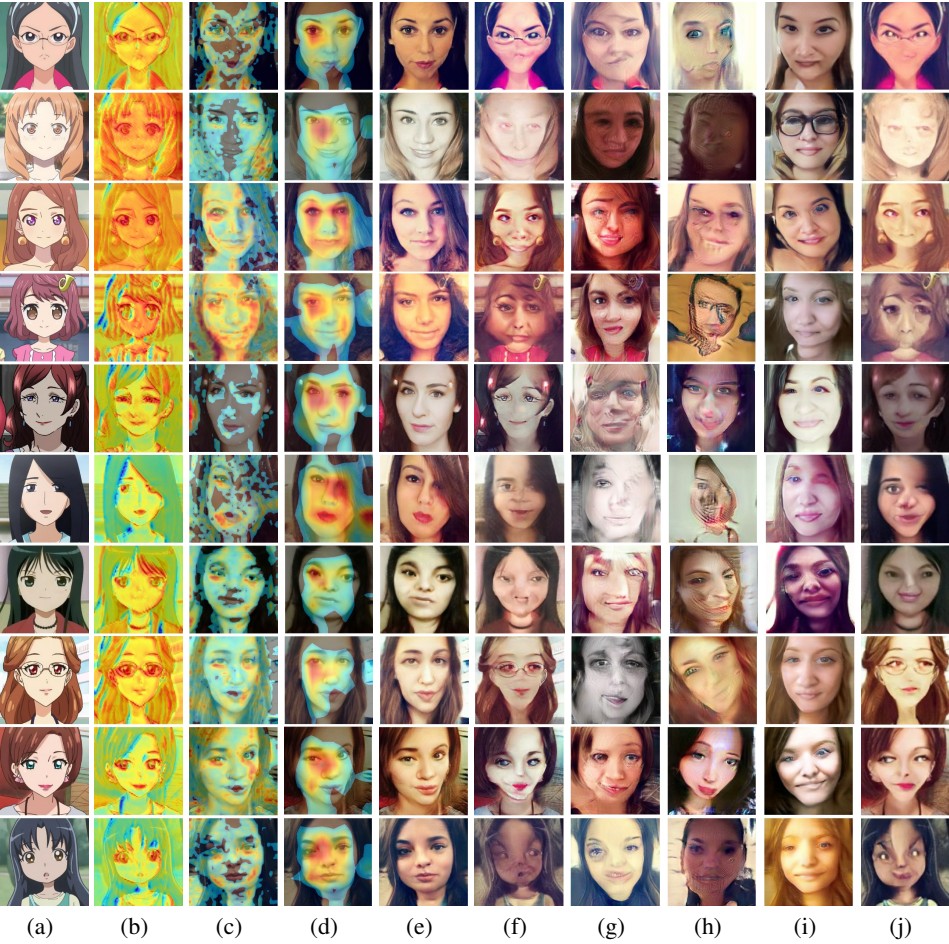

(a)  (b)  (c)  (d)  (e)  (f)  (g)  (h)  (i)  (j)

Figure 6: Visual comparisons of the anime2selfie with attention features maps. (a) Source images, (b) Attention map of the generator, (c-d) Local and global attention maps of the discriminators, (e) Our results, (f) CycleGAN (Zhu et al. (2017)), (g) UNIT (Liu et al. (2017)), (h) MUNIT (Huang et al. (2018)), (i) DRIT (Lee et al. (2018)), (j) AGGAN (Mejjati et al. (2018)).

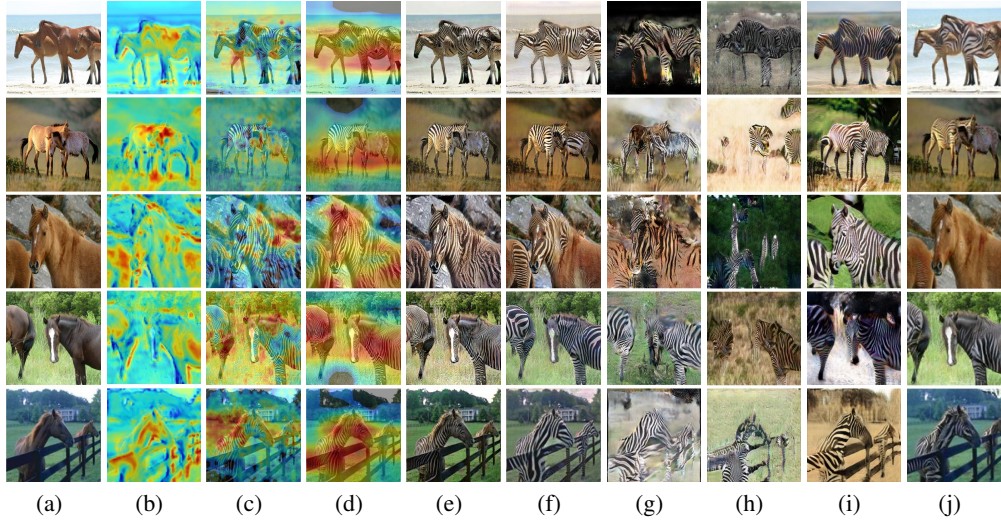

Figure 7: Visual comparisons of the horse2zebra with attention features maps. (a) Source images, (b) Attention map of the generator, (c-d) Local and global attention maps of the discriminators, (e) Our results, (f) CycleGAN (Zhu et al. (2017)), (g) UNIT (Liu et al. (2017)), (h) MUNIT (Huang et al. (2018)), (i) DRIT (Lee et al. (2018)), (j) AGGAN (Mejjati et al. (2018)).

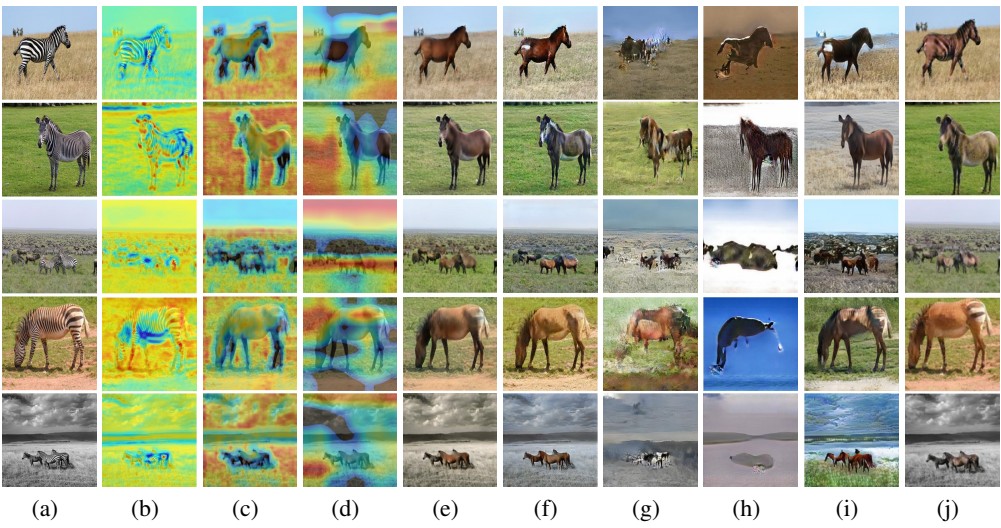

Figure 8: Visual comparisons of the zebra2horse with attention features maps. (a) Source images, (b) Attention map of the generator, (c-d) Local and global attention maps of the discriminators, (e) Our results, (f) CycleGAN (Zhu et al. (2017)), (g) UNIT (Liu et al. (2017)), (h) MUNIT (Huang et al. (2018)), (i) DRIT (Lee et al. (2018)), (j) AGGAN (Mejjati et al. (2018)).

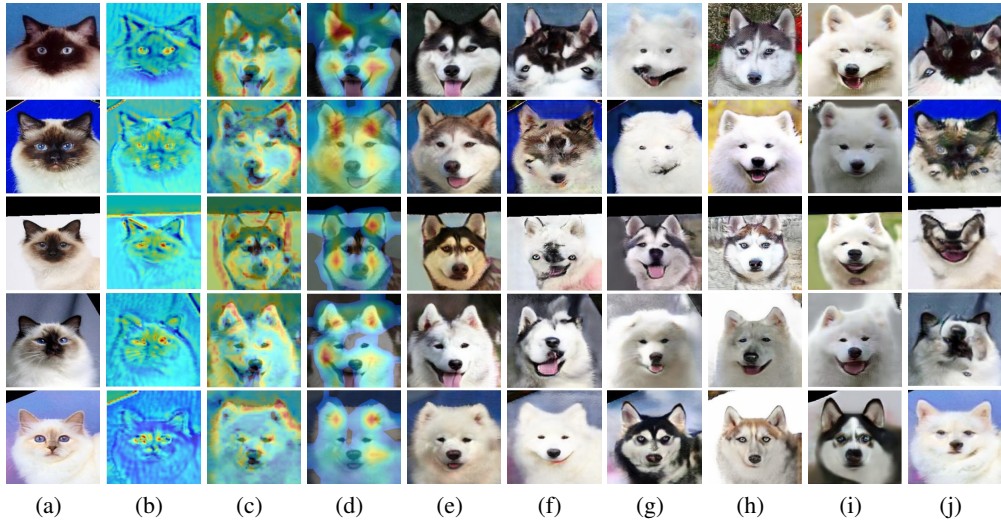

Figure 9: Visual comparisons of the cat2dog with attention features maps. (a) Source images, (b) Attention map of the generation, (c-d) Local and global attention maps of the discriminators, (e) Our results, (f) CycleGAN (Zhu et al. (2017)), (g) UNIT (Liu et al. (2017)), (h) MUNIT (Huang et al. (2018)), (i) DRIT (Lee et al. (2018)), (j) AGGAN (Mejjati et al. (2018)).

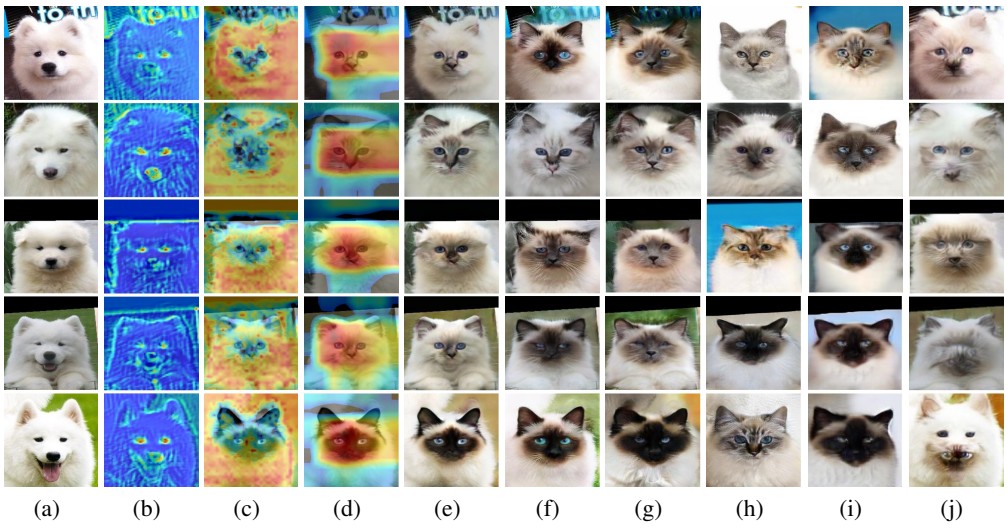

Figure 10: Visual comparisons of the dog2cat with attention features maps. (a) Source images, (b) Attention map of the generation, (c-d) Local and global attention maps of the discriminators, (e) Our results, (f) CycleGAN (Zhu et al. (2017)), (g) UNIT (Liu et al. (2017)), (h) MUNIT (Huang et al. (2018)), (i) DRIT (Lee et al. (2018)), (j) AGGAN (Mejjati et al. (2018)).

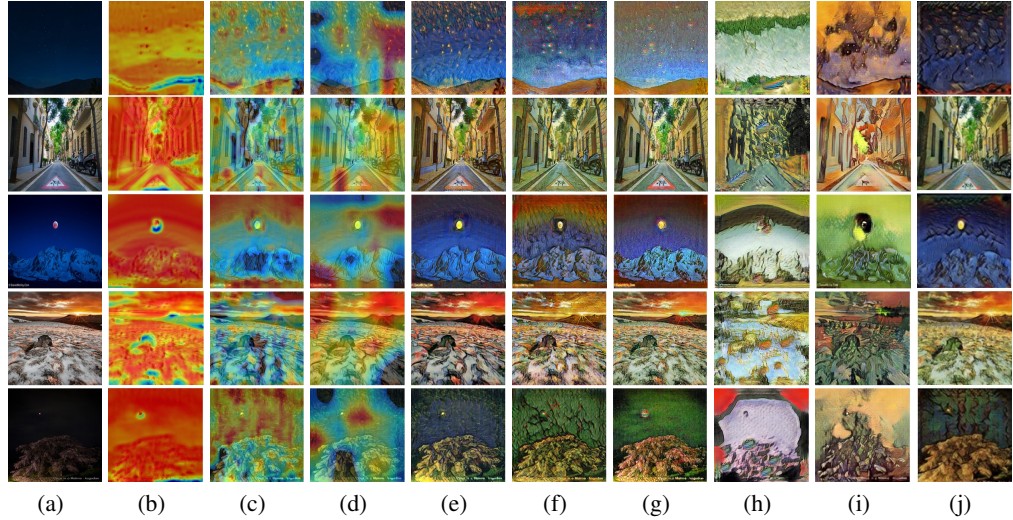

(a)  (b)  (c)  (d)  (e)  (f)  (g)  (h)  (i)  (j)

Figure 11: Visual comparisons of the photo2vangogh with attention features maps. (a) Source images, (b) Attention map of the generation, (c-d) Local and global attention maps of the discriminators, respectively, (e) Our results, (f) CycleGAN (Zhu et al. (2017)), (g) UNIT (Liu et al. (2017)), (h) MUNIT (Huang et al. (2018)), (i) DRIT (Lee et al. (2018)), (j) AGGAN (Mejjati et al. (2018)).

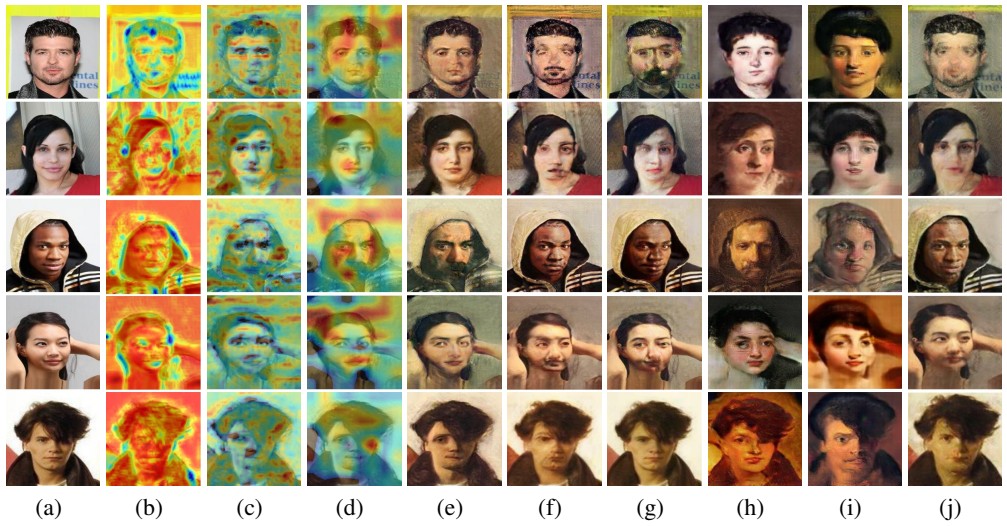

(a)  (b)  (c)  (d)  (e)  (f)  (g)  (h)  (i)  (j)

Figure 12: Visual comparisons of the photo2portrait with attention features maps. (a) Source images, (b) Attention map of the generator, (c-d) Local and global attention maps of the discriminators, respectively, (e) Our results,(f) CycleGAN (Zhu et al. (2017)), (g) UNIT (Liu et al. (2017)), (h) MUNIT (Huang et al. (2018)), (i) DRIT (Lee et al. (2018)), (j) AGGAN (Mejjati et al. (2018)).

