# OpenReview forum: "U-GAT-IT: Unsupervised Generative Attentional Networks with Adaptive Layer-Instance Normalization for Image-to-Image Translation"
_ICLR.cc/2020/Conference — Accept (Poster)_

### Official Review · AnonReviewer3 · 2019-10-22
**Official Blind Review #3**

**Rating:** 6

**Review:**

This paper proposes a new attention mechanism for unsupervised image-to-image translation task. The proposed attention mechanism consists of an attention module and a learnable normalization function. Sufficient experiments and analysis are done on five datasets.

Pros:
1. The proposed method seems to generalize well to the different datasets with the same network architecture and hyper-parameters compared to previous works. This could benefit other researchers who want to apply the method to other data or tasks.
2. The translated results seem more semantic consistent with the source image compared to other methods, although the sores are not the top on photo2portrait and photo2vangogh. The results also look more pleasing.

Cons:
1. The CAM loss is one of the key components in the proposed method. However, there is only the reference and no detailed description in the paper. More intuitive descriptions are necessary for easy understanding.
2. The local and global discriminators are not explained until the result analysis. It’s a bit confusing when I see the local and global attention maps visualization results. It’s better to mention it in the method section.
3. I wonder why some translations are not done at all in the results without CAM in Figure 2(f). Because without CAM, the framework would be somehow similar to MUNIT or DRIT. I suppose the hyper-parameters are not suitable for this setting.
4. The generator model architecture in Figure 1 is confusing. The adaptive residual blocks only receive the gamma and beta parameters. I suppose that the encoder feature maps are also fed into the adaptive residual blocks.
5. In Figure 3, the comparison of the results using each normalization function is reported. While in my view, the results only using GN in decoder with CAM looks more natural. I wonder why the proposed method only consists of instance norm and layer norm? I suppose the group norm might help with the predefined group.
6. In the ablation study, the CAM is evaluated for generator and discriminator together. I would recommend doing this ablation study for generator and discriminator separately to see if it’s necessary for generator or discriminator.
7. It would be good to see some discussion on the attention mechanism compared with other related works. For example,  [a,b] predict the attention masks for unsupervised I2I, but applies them on the pixel/feature spatial level to keep the semantic consistency.
[a] Unsupervised-Attention-guided-Image-to-Image-Translation. NIPS’18
[a] Exemplar guided unsupervised image-to-image translation with semantic consistency. ICLR’19

My initial rating is above boardline.

**Experience Assessment:**

I have published one or two papers in this area.

**Review Assessment: Checking Correctness Of Derivations And Theory:**

I assessed the sensibility of the derivations and theory.

**Review Assessment: Checking Correctness Of Experiments:**

I carefully checked the experiments.

**Review Assessment: Thoroughness In Paper Reading:**

I read the paper at least twice and used my best judgement in assessing the paper.

---

> ### Author Response · Authors · 2019-11-08
> **Response to Reviewer #3**
>
> Thank you for the valuable comments and constructive feedback.  In the revised draft, we mark our major revisions by “violet” , and would like to answer the reviewer’s questions as follows:
>
> 1. Description for CAM
>
> As you suggested, in the revised draft, we add the related works including the description for CAM in Appendix A.
>
> 2. The local and global discriminators
>
> As you suggested, in the revised draft, we add the description for the multi-scale discriminator in Sec. 2.1.2.
>
> 3. Result without CAM (Figure2(f))
>
> To make sure that the effect of the CAM, we have all set the same hyper-parameters and retrained the model.
>
> 4. The generator model architecture  (Figure 1)
>
> You're right. We will modify the figure to make it appear that the encoder feature maps are fed into the adaptive residual blocks.
>
>
> 5. Why not using GN?
>
> We thought GN was theoretically an intermediate version of IN and LN. Therefore, the GN can be properly expressed by the /rho value. The selection was based on whether the result would be closer to the target domain or more biased toward the source domain rather than the naturality of the results. In Figure 3 (f), you can see more textures from the background of the source domain.
>
> 6. Ablation Study
>
> As shown in Table 1, it includes the ablation study for generator and discriminator separately with KID. If space is allowed, we will add the image results.
>
> 7. Discussion on the attention mechanism compared with other related works
>
> Our experiment results already are including the result of AGGAN[1] and discussed about that.
> [1] Unsupervised-Attention-guided-Image-to-Image-Translation. NIPS’18

---

### Official Review · AnonReviewer1 · 2019-10-23
**Official Blind Review #1**

**Rating:** 8

**Review:**

I have read the authors' rebuttal and satisfied with their response. Novelty is a little on the lower side, but thorough writing, results, and insightful comparisons make up for this in my opinion. I have updated my score to 8: Accept.

=====

This paper proposes an approach to perform image translation called U-GAT-IT. In image translation, the goal is to learn a mapping from images in a source domain to corresponding images in a target domain. Contemporary image translation approaches are able to transfer local texture but struggle to handle shape transfer. To address this concern, the authors introduce an attention mechanism based on CAM [1] and an adaptive normalization layer into a GAN-based image translation framework. Results indicate favorable quantitative and qualitative performance relative to a number of baselines.

Specific contributions include:
* Introduction of a normalization layer called AdaLIN that can interpolate between instance normalization and layer normalization based on the input.
* Introduction of an attention mechanism based on CAM [1] that allows the model to focus on specific parts of the image when either generating or discriminating.
* Collection and release of a selfie-to-anime dataset.
* Release of U-GAT-IT code.

In my opinion this paper is borderline, leaning towards weak accept. The experiments are thorough and the paper is well-written. I have concerns about the novelty and significance of the work, but overall the paper feels very close to being a finished piece of work in spite of its (relatively minor) flaws.

Strong points of this work include the writing and experiments. The paper is clearly organized and feels polished. It cites many relevant works, giving the reader a sense of the contemporary approaches for image translation. There is a thorough description of model architecture, dataset and tuning parameters in the appendix. In addition, code and the selfie-to-anime dataset have been released by the authors. In terms of experiments, the authors provide many qualitative visualizations comparing the proposed model to baselines on various datasets. Quantitative evaluation includes KID and a perceptual evaluation on human subjects.

Weak points include novelty and significance. The proposed approach combines two ideas already applied to image translation (adaptive normalization [3] and attention [4]).  It therefore synthesizes these ideas into an effective algorithm rather than directly adding something new. It is unclear to me how others can build on top of this work to further advance state-of-the-art in image translation. Are more sophisticated normalization and attention mechanisms truly the key to improving image translation in the future?

Specific comments:
* The formulation of AdaLIN in Equation (1) is vague. The text states "parameters are dynamically computed by a fully connected layer from the attention map", but it's not clear what those parameters are in the equation. Explicitly writing \gamma and \beta as functions of the fully-connected layer and \mu_I, \sigma_I, \mu_L, \sigma_L as the corresponding mean and standard deviation expressions would make things more clear.
* The motivation for using layer normalization was discussed in 2.1.1 but I still do not understand why it is beneficial.
* The term "importance weights" has a specific meaning in the context of Monte Carlo methods. I would suggest choosing a different term here.

Questions for the authors:
* How does U-GAT-IT compare to TransGaGa [2]? One of the stated goals of U-GAT-IT is to better handle shape when performing image translation. TransGaGa has a similar motivation and so I would have liked to see an experimental comparison or at the very least a description of how U-GAT-IT differs. What sorts of shape transfer could U-GAT-IT handle that TransGaGa couldn't and vice versa?
* What are the shortcomings of the model and how could they possibly be addressed?

[1] Zhou, B., Khosla, A., Lapedriza, A., Oliva, A. and Torralba, A., 2016. Learning deep features for discriminative localization. In Proceedings of the IEEE conference on computer vision and pattern recognition (pp. 2921-2929).
[2] Wu, W., Cao, K., Li, C., Qian, C. and Loy, C.C., 2019. Transgaga: Geometry-aware unsupervised image-to-image translation. In Proceedings of the IEEE Conference on Computer Vision and Pattern Recognition (pp. 8012-8021).
[3] Huang, X. and Belongie, S., 2017. Arbitrary style transfer in real-time with adaptive instance normalization. In Proceedings of the IEEE International Conference on Computer Vision (pp. 1501-1510).
[4] Mejjati, Y.A., Richardt, C., Tompkin, J., Cosker, D. and Kim, K.I., 2018. Unsupervised attention-guided image-to-image translation. In Advances in Neural Information Processing Systems (pp. 3693-3703).

**Experience Assessment:**

I do not know much about this area.

**Review Assessment: Checking Correctness Of Derivations And Theory:**

I assessed the sensibility of the derivations and theory.

**Review Assessment: Checking Correctness Of Experiments:**

I assessed the sensibility of the experiments.

**Review Assessment: Thoroughness In Paper Reading:**

I read the paper at least twice and used my best judgement in assessing the paper.

---

> ### Author Response · Authors · 2019-11-07
> **Response to Reviewer #1**
>
> We thank the reviewer for the valuable comments and constructive feedback.  In the revised draft, we mark our major revisions by “violet” , and would like to answer the reviewer’s questions as follows:
>
> *About Novelty:
>
> 1. Attention mechanism
>
> Although we proposed similar attention concept, the goal and how to generate are different. Previous attention-based work [1] do not allow to transform the shape of the instance because of attaching the background to the (translated) cropped instances. Unlike these works, we assumed that our model will guide to focus on more important regions and ignore minor regions by distinguishing between the target and not-target domains based on the importance map obtained by the auxiliary classifier. These attention maps are embedded into the generator and discriminator to focus on semantically important areas, thus facilitating the shape transformation. The attention map in the generator induces focus on areas that specifically distinguish between the two domains. The attention map in discriminator helps fine-tuning by focusing on the difference between real image and fake image in target domain.
>
> 2. Normalization
>
> AdaLIN tells the model how much it should transform. Instance Norm (IN) is capable of preserving the characteristics of source image. Layer Norm (LN) which uses layer-wise feature statistics is better at transforming to target domain. We found that combining advantages of both IN and LN is beneficial to image-to-image translation task in various datasets by controlling the amount of transform. Though its idea borrows from previous work [2], the proposed method is the first attempt to combine IN and LN in image-to-image translation task as far as we investigated.
>
> [1] Y. Alami Mejjati, C. Richardt, J. Tompkin, D. Cosker, and K. I. Kim. Unsupervised attention-guided image-to-image translation. In NIPS. 2018.
> [2] H. Nam and H.-E. Kim. Batch-instance normalization for adaptively style-invariant neural networks. In NIPS, 2018.
>
> *Specific Comments:
>
> 1. The formulation of AdaLIN in Equation (1)
>
> I agree with the comment from the Reviewer. In the revised draft, I modify like  "parameters, /gamma and /beta are dynamically computed by a fully connected layer from the attention map" in sec 2.1.1.
>
> 2.  The motivation for using layer normalization
>
> We assumed that optimal stylization method was "whitening and Coloring Transform". However, the computational cost is high due to the calculation of the covariance matrix and matrix inverse. To compensate between the computational cost and the quality of the result, we borrowed two sub-optimal normalization methods, AdaIN and Layer Normalization. During stylization, while the AdaIN has the characteristics keeping more contents information, the Layer Normalization tends to make stylization more obvious instead keeping content information less.
>
> 3. The term "important weights"
>
> In the revised draft, we modify it to "the weight of the k-th feature map for the source domain"  in sec 2.1.1.
>
> *Questions for the authors:
>
> 1. U-GAT-IT vs TransGaGa
>
> The goal of U-GAT-IT is to change the shape of the foreground while maintaining the content of the background. TransGaGa deal with the geometry and appearance separately and fully converts the appearance of the source domain into that of the target domain without considering the foreground and background. Therefore, as can be seen from the experimental results, the background of the source image is not maintained at all. However, U-GAT-IT can maintain or change the contents of the source domain adaptively through attention. In addition, U-GAT-IT can achieve good results in style transfer as well as shape change through AdaLIN. Therefore, we think U-GAT-IT is a more generalized version than TransGaGa.
>
> 2. What are the shortcomings of the model and how could they possibly be addressed?
>
> The shortcomings for our model is "one-to-one mapping". But we will design UGATIT with our future work to be multi-modal and multi-domain together.

---

### Official Review · AnonReviewer2 · 2019-10-27
**Official Blind Review #2**

**Rating:** 6

**Review:**

Summarize what the paper claims to do/contribute.
* The paper proposes a new image-to-image GAN-based translator that uses attention and a new normalization that learns a proper ratio between instance and layer normalization. Experiments benchmark the new method against multiple prior ones, and on a number of dataset pairs.

Clearly state your decision (accept or reject) with one or two key reasons for this choice.
Weak Accept

* The paper was well-written and the method and contributions are clearly explained.
* There is clear novelty in this paper, even if slightly limited. However, the newly proposed normalization seems to work quite well.
* The results look good, however it is hard to compare methods quantitatively with only few samples. (Nothing that the authors could have done: there are many samples in the supplementary material and results seem consistent.) Qualitative measures like FID and KID should be taken with a grain of salt also. It is a big plus that a user study was conducted! (However, details of how these subjects were selected would be useful)

**Experience Assessment:**

I have published one or two papers in this area.

**Review Assessment: Checking Correctness Of Derivations And Theory:**

I assessed the sensibility of the derivations and theory.

**Review Assessment: Checking Correctness Of Experiments:**

I carefully checked the experiments.

**Review Assessment: Thoroughness In Paper Reading:**

I made a quick assessment of this paper.

---

> ### Author Response · Authors · 2019-11-07
> **Response to Reviewer #2**
>
> We thank the reviewer for the valuable comments and constructive feedback , and would like to answer the reviewer’s questions as follows:
> The perceptual study was conducted on 153 participants of an AI community, which includes a mix of experts and non-specialists.

---

### Public Comment · ~Kyunghyun_Cho1 · 2019-10-14
**[REDACTED BY PROGRAM CHAIR] Double-blind review is violated**

[NOTE BY PROGRAM CHAIR: the urls were redacted to preserve the anonymity. this submission does *not* violate the ICLR's policy on double blind reviewing. see https://iclr.cc/Conferences/2020/CallForPapers which states "... papers that have appeared on non-peered reviewed websites (like arXiv) or that have been presented at workshops (i.e., venues that do not have a publication proceedings) do not violate the policy. The policy is enforced during the whole reviewing process period. Submission of the paper to archival repositories such as arXiv are allowed."]

It seems like double-blind review is violated. This paper was published a long time ago (25 July 2019).

[URL REDACTED]

There are all author names at the top of the paper, their affiliation, etc.
Furthermore, there is an official implementation of this model on GitHub for both Tensorflow (4k stars) and Pytorch (1.3k stars) frameworks of the first author of the paper [REDACTED] and his co-author [REDACTED].

[URL REDACTED]
[URL REDACTED]

---

### Decision · Program_Chairs · 2019-12-19

**Decision:**

Accept (Poster)

**Comment:**

The paper proposes a new architecture for unsupervised image2image translation.
Following the revision/discussion, all reviewers agree that the proposed ideas are reasonable, well described, convincingly validated, and of clear though limited novelty. Accept.